# Supplemental Far-Red Light Stimulates Lettuce Growth: Disentangling Morphological and Physiological Effects

**DOI:** 10.3390/plants10010166

**Published:** 2021-01-16

**Authors:** Reeve Legendre, Marc W. van Iersel

**Affiliations:** Department of Horticulture, University of Georgia, Athens, GA 30602, USA; reeve.legendre@uga.edu

**Keywords:** canopy size, incident light, leaf morphology, light-emitting diodes, light interception, light use efficiency, radiation use efficiency

## Abstract

Light-emitting diodes allow for the application of specific wavelengths of light to induce various morphological and physiological responses. In lettuce (*Lactuca sativa*), far-red light (700–800 nm) is integral to initiating shade responses which can increase plant growth. In the first of two studies, plants were grown with a similar photosynthetic photon flux density (PPFD) but different intensities of far-red light. The second study used perpendicular gradients of far-red light and PPFD, allowing for examination of interactive effects. The far-red gradient study revealed that increasing supplemental far-red light increased leaf length and width, which was associated with increased projected canopy size (PCS). The higher PCS was associated with increased cumulative incident light received by plants, which increased dry matter accumulation. In the perpendicular gradient study, far-red light was 57% and 183% more effective at increasing the amount of light received by the plant, as well as 92.5% and 162% more effective at increasing plant biomass at the early and late harvests, respectively, as compared to PPFD. Light use efficiency (LUE, biomass/mol incident light) was generally negatively correlated with specific leaf area (SLA). Far-red light provided by LEDs increases the canopy size to capture more light to drive photosynthesis and shows promise for inclusion in the growth light spectrum for lettuce under sole-source lighting.

## 1. Introduction

Light-emitting diodes (LEDs) are a rapidly advancing lighting technology in controlled environment agriculture. LED fixtures can be used for supplemental lighting in greenhouses or for sole-source lighting in plant factories. The decreasing cost and increasing efficacy of LED fixtures, coupled with the flexibility to provide different wavebands and control light intensity, has increased their market share. The worldwide predicted 2021 market value of LED fixtures used in controlled environment agriculture is USD 1,800,000,000, a 350% increase from 2014 [1]. The growth of the market is unsurprising, as LEDs provide many advantages compared to older lighting technologies, such as high-pressure sodium (HPS) or fluorescent lamps. LED fixtures are smaller, require less maintenance, have a longer life, and are more efficient [2]. Part of the reason for the higher efficiency is that LED fixtures operate at much cooler temperatures than older technologies, losing less energy to the generation of heat. The lower temperature and lack of radiant heat allows for LED fixtures to be positioned much closer to the plant canopy [3,4]. The efficiency of LED fixtures can be further increased by generating light only in the photosynthetically active radiation (PAR) range that drives photosynthesis [5]. The ability of LED fixtures to provide specific spectra of light can be used to illicit morphological responses, such as flowering, shade avoidance pathways, alteration of metabolomes, and increasing plant defenses, plant size, and growth rates [6,7,8,9,10]. LEDs can even have utility after harvest as postharvest application of different light spectra can influence volatile flower and fruit compounds [11].

Far-red radiation (FR, 700–800 nm) shows promise for inclusion in the spectrum of LED grow lights. As plants are immobile, they need to adapt to their environment, which occurs partly by detecting and responding to light quality. Far-red radiation is absorbed poorly by leaves compared to PAR (400–700 nm), lowering the red to far-red ratio (R:FR) in the shade under the canopy of plants. A decrease in the R:FR ratio can elicit a shade-avoidance or shade-tolerance response, because of the ability of plants to detect changes in the R:FR ratio via phytochromes. When exposed to a low R:FR ratio, shade-tolerant species often display thinner, larger leaves with a higher chlorophyll content to capture more light [12]. Under the same conditions, shade-avoidant species often display faster elongation of stems and leaves to break through the canopy to reach more light. The plants may also have lower chlorophyll content and strong apical dominance [13].

In addition to initiating morphological responses, far-red radiation can increase photosynthesis. While PAR is typically defined as the wavelengths from 400 to 700 nm [14], far-red light increases photosynthesis in a synergistic manner with light of shorter wavelengths. First described as the Emerson enhancement effect, simultaneous application of red and far-red light increases photosynthetic rates when compared to independent application of those light spectra [15,16]. The quantum yield of photosystem II (Φ_PSII_) and net photosynthetic rate of lettuce (*Lactuca sativa*) increase when far-red light is added to white or red and blue light by preferentially exciting photosystem I, which may be under-excited by light spectra without far-red [17]. Zhen and Bugbee [18] recently demonstrated that far-red photons increase canopy photosynthesis as effectively as 400–700 nm photons.

Lettuce is an economically important crop which is often grown in greenhouses with supplemental lighting and is the second-most consumed vegetable crop in the US at 11.7 kg per person per year. The market value of lettuce production in the US was around USD 1,900,000,000 in 2017 [19]. Some lettuce varieties exhibit shade tolerance responses in response to supplemental far-red light, such as increased leaf expansion and seedling growth [20,21]. Far-red light treatments can also increase fresh and dry weight, leaf expansion, leaf area, shoot height and internodal length, accumulation of bioactive compounds, and mitotic cell division in lettuce [22,23,24,25].

Although supplemental far-red light can increase the growth of lettuce, it currently is not clear whether far-red light stimulates growth more than adding the same amount of PAR photons. Quantifying the relative effects of PAR and far-red light is necessary to determine whether far-red LEDs should be included in LED fixtures. This requires a better understanding of the underlying mechanisms by which far-red light can increase growth, likely a combination of morphological and physiological effects. To separate these effects, plant growth can be separated into two components: the amount of incident light on the canopy (a function of morphology, specifically the canopy size, and light intensity) and the efficiency with which that incident light is used to produce biomass (a physiological measure). Spectral effects on light use efficiency (LUE; g of biomass/mol of light) of lettuce have been studied previously, with a 8:1:1 ratio of red, green, and blue light resulting in the highest LUE [26]. The addition of far-red light increased the LUE by enlarging the leaf area, allowing plants to capture more light [27]. However, these previous studies calculated LUE based on the total amount of light provided to the entire growing area, which does not allow for a separation of the morphological and physiological effects of far-red light on plant growth. Calculating LUE based on the amount of light reaching the canopy, rather than the entire growing area, does allow for such a separation. 

The shade-tolerance response of developing larger leaves increases canopy size and thus incident light, potentially leading to increased canopy photosynthesis and growth. Projected canopy size (PCS), acquired through imaging, is a nondestructive measurement that can be taken throughout the growing period to quantify canopy growth. Projected canopy size is a good indicator of how much light is absorbed by the crop and is strongly correlated with daily carbon gain, a direct measure of growth [28]. Nondestructive canopy size measurements, such as PCS, combined with photosynthetic photon flux density (PPFD, 400–700 nm) and far-red intensities, can be used to quantify the incident light that a plant receives and to estimate the light use efficiency (LUE; g of biomass/mol of incident light). Thus, canopy imaging, combined with dry mass data, can be used to simply quantify morphological and physiological effects of light spectra that together determine growth: how much light reaches the canopy and how efficiently this incident light is used to produce biomass. 

We conducted two studies to quantify the effects of far-red light on the growth of lettuce. The first study used a far-red gradient to determine the effect of supplemental far-red light over a narrow PPFD range. The goal was to quantify the effects of far-red light on lettuce morphology, LUE, and growth. We hypothesized that supplemental far-red would induce an intensity-dependent shade tolerance response, increasing the PCS and, consequently, the incident light, leading to more growth. The second study used perpendicular gradients of far-red light and PPFD to evaluate the interactive effects of far-red light and PPFD and to quantify their relative efficacy in increasing growth. We hypothesized that far-red light would increase canopy size more than PPFD, because of the shade-tolerance response induced by far-red light. In addition, we hypothesized that far-red supplementation would not decrease the LUE, regardless of PPFD level, because far-red light can drive photosynthesis as effectively as PPFD [18]. We hypothesized that far-red light would be more effective than PPFD in increasing growth because it can stimulate leaf elongation, increase PCS and incident light, and drive photosynthesis.

## 2. Results

### 2.1. Far-Red Gradient Experiment

In the first experiment, a far-red gradient (4.9–28.0 µmol m^−2^ s^−1^) was established using far-red LEDs (peak at 735 nm) in a growth chamber. A narrow range PPFD (207 ± 13 µmol m^−2^ s^−1^) was provided using cool-white LED panels to minimize variability in PPFD. Sixty “Green Salad Bowl” lettuce plants were grown, with half harvested at 16 days after germination (early), while the remainder were harvested at 25 days after germination (late). The objective of this study was to quantify the effects of different intensities of supplemental far-red light on the morphology, LUE, and growth of “Green Salad Bowl” lettuce. Statistical analyses indicated no significant effects of differences in PPFD on plant morphological or physiological responses. 

#### 2.1.1. Leaf Morphology

The supplemental far-red light affected leaf morphology in both the early and late harvests. At both harvests, increasing the intensity of supplemental far-red light increased leaf length and width (*p* ≤ 0.014; Figure 1). At the early harvest, leaf length increased by 0.99 mm per µmol m^−2^ s^−1^ of far-red light, while leaf width increased by 0.75 mm per µmol m^−2^ s^−1^. In the late harvest, leaf length increased by 1.1 mm per µmol m^−2^ s^−1^, while leaf width increased by 0.83 mm per µmol m^−2^ s^−1^.

#### 2.1.2. Canopy Traits

Projected canopy size was measured nondestructively using an imaging system and is a measure of how much of the canopy is exposed to direct light. Leaf area was measured on the combined leaves of each plant at both harvests. Leaf length and width were positively correlated with both leaf area and PCS in both harvests (*p <* 0.0001; Figure 2). An increase of 1 cm in leaf length was associated with a 49.7 cm^2^/plant increase in leaf area in the early harvest and 101.4 cm^2^/plant in the late harvest, while a 1 cm increase in leaf width was associated with an increase in leaf area of 40.8 cm^2^/plant in the early harvest and 99.0 cm^2^ in the late harvest. Projected canopy size increased by 38.0 cm^2^/plant at the early harvest and 44.1 cm^2^/plant at the late harvest for every centimeter increase in leaf length and by 30.1 cm^2^ at the early harvest and 32.3 cm^2^ at the late harvest per cm of leaf width (Figure 2).

#### 2.1.3. Incident Light

The PCS of each plant was estimated for each day using regression analysis. When combined with the instantaneous PPFD and supplemental far-red levels from wavelengths from 400 to 800 nm, the daily incident light was estimated. The daily incident light was integrated to estimate the cumulative incident light over the course of the experiment for each plant. At both harvests, both leaf area and PCS were positively correlated with incident light (Figure 3). The cumulative incident light was positively correlated with plant dry weight in both harvests (*p* < 0.0001; Figure 3C). Each 1 m^2^ increase in leaf area was associated with a 47 and 59 mol increase in incident light up to the early and late harvests, respectively. Each 1 m^2^ increase in PCS was associated with an increase in incident light by 62 and 146 mol at the early and the late harvests, respectively. Shoot dry weight increased by 0.552 and 0.506 g per mol of incident light at the early and late harvests, respectively.

At the early harvest, increasing the intensity of supplemental far-red light increased the amount of cumulative incident light (400–800 nm) by 14 (mmol/plant)/(µmol m^−2^ s^−1^) (*p* = 0.0006; Figure 4A) and plant dry weight by 32 (mg/plant)/(µmol m^−2^ s^−1^) (*p* = 0.0048; Figure 4B). Likewise, increasing supplemental far-red light increased the cumulative amount of incident light by 71 (mmol/plant)/(µmol m^−2^ s^−1^) (*p* < 0.0001; Figure 4A) and the dry weight by 145 (mg/plant)/(µmol m^−2^ s^−1^) at the late harvest (*p* < 0.0001; Figure 4B).

#### 2.1.4. Light Use Efficiency

Light use efficiency (shoot dry mass per mol of incident light) was calculated in two ways: by using only the PPFD wavelengths (LUE_PPFD_; 400–700 nm) or by using the total photon flux, which includes the far-red wavelengths (LUE_TOTAL_; 400–800 nm). The LUE_PPFD_ ranged from 0.568 to 0.828 g mol^−1^, while the LUE_TOTAL_ ranged from 0.518 to 0.806 g mol^−1^. The plants from the early harvest had a higher LUE_TOTAL_ and LUE_PPFD_ than the plants from the final harvest. While far-red light caused no change in the LUE_PPFD_, increasing supplemental far-red slightly decreased the LUE_TOTAL_ (Figure 5). The LUE_TOTAL_ was negatively correlated with SLA in both the early (*p* = 0.014) and late harvests (*p* = 0.029), while the LUE_PPFD_ was negatively correlated with SLA at only the early harvest (*p* = 0.014).

### 2.2. Perpendicular Light Gradient Experiment

In the perpendicular light gradient experiment, a PPFD gradient (111–245 µmol m^−2^ s^−1^) was established orthogonally to a far-red gradient (4.7–32.8 µmol m^−2^ s^−1^) to achieve a range of combinations of high and low PPFD levels with high and low amounts of far-red light. The objective of the study was to quantify potential interactions between far-red light and PPFD and their relative efficacy in stimulating biomass production of lettuce “Green Salad Bowl”. There were no interactions between PPFD and far-red throughout the experiment, so we focus on the main effects of far-red light and PPFD.

#### 2.2.1. Leaf Morphology

While far-red had similar effects on leaf morphology as in the far-red gradient experiment, PPFD also altered leaf morphology. In the early harvest, leaf length increased with increasing far-red (*p* < 0.0001), while increasing PPFD tended to decrease the length of leaves (Figure 6A). At the late harvest, the antagonistic effect of far-red and PPFD was still present, with leaf length increasing with increasing far-red (*p* < 0.005) but decreasing with increasing PPFD (*p* = 0.003; Figure 6B). Leaf width at the early harvest was increased with both increasing far-red (*p* = 0.001) and PPFD (*p* = 0.02; Figure 6C). At the late harvest, higher far-red increased leaf width (Figure 6D), but leaf width was not affected by PPFD.

#### 2.2.2. Canopy Traits

While leaf length was not correlated with either PCS or leaf area (results not shown), leaf width was positively correlated with leaf area and PCS at both harvests (Figure 7). Each 1 cm increase in leaf width was associated with a 40.3 cm^2^ increase in leaf area at the early harvest and a 51.7 cm^2^ increase at the late harvest. Each 1 cm increase in leaf width was associated with a 43.4 cm^2^ increase in PCS at the early harvest and 40.0 cm^2^ at the late harvest.

Larger leaf areas and PCS resulted in more cumulative incident light at both the early and late harvests. The amount of cumulative incident light was positively correlated with shoot dry weight at both harvests (Figure 8). An increase in the leaf area was associated with increasing cumulative incident light by 29 mol m^−2^ at the early harvest and 70 mol m^−2^ at the late harvest. A 1 m^2^ increase in PCS was associated with increasing cumulative incident light by 28 mol/plant and 86 mol/plant the early and late harvest, respectively. Shoot dry weight increased by 0.796 and 0.476 g per mole incident light at the early and late harvest, respectively.

Dry weight was positively correlated with both far-red intensity and PPFD at both harvests (Figure 9; Table 1). One of our main objectives was to quantify the efficacy with which far-red light and PPFD increase incident light and biomass. The cumulative incident light from germination to the early harvest increased by 8.15 (mmol/plant)/(µmol m^−2^ s^−1^) for far-red light and 5.20 (mmol/plant)/(µmol m^−2^ s^−1^) for PPFD, making far-red light 57% more effective than PPFD at increasing cumulative incident light. From germination until the late harvest, increasing PPFD increased cumulative incident light by 28.1 (mmol/plant)/(µmol m^−2^ s^−1^), while far-red light increased cumulative incident light by 79.4 (mmol/plant)/(µmol m^−2^ s^−1^), making far-red 183% more effective at increasing cumulative incident light. Far-red light was also more efficient at increasing dry weight than PPFD. At the early harvest, far-red light increased biomass by 7.92 (mg/plant)/(µmol m^−2^ s^−1^) compared to 4.12 (mg/plant)/(µmol m^−2^ s^−1^) for PPFD, making far-red 92.5% more effective than PPFD at increasing dry weight. At the late harvest, far-red light was associated with an increase in dry weight by 42.76 (mg/plant)/(µmol m^−2^ s^−1^), while PPFD increased dry weight by 11.83 (mg/plant)/(µmol m^−2^ s^−1^). Far-red light was 162% more effective at increasing the weight of the plant at the late harvest.

The LUE_TOTAL_ ranged from 0.503 to 0.838 g mol^−1^ at the early harvest and 0.258 to 0.582 g mol^−1^ at the late harvest, while the LUE_PPFD_ ranged from 0.525 to 0.912 g mol^−1^ at the early harvest and 0.276 to 0.674 g mol^−1^ at the late harvest. At the early harvest, both LUE_TOTAL_ and LUE_PPFD_ were higher than at the late harvest (*p <* 0.0001). The LUE_PPFD_ was positively correlated with far-red levels (Figure 10) but was unaffected by PPFD. The LUE_TOTAL_ was not correlated with either far-red light intensity or PPFD. Both the LUE_TOTAL_ and LUE_PPFD_ were negatively correlated with SLA at the early and late harvests (Figure 11).

## 3. Discussion

### 3.1. Far-Red Light and PPFD Change Leaf Morphology

In both the far-red light gradient experiment and the perpendicular light gradient experiment, supplemental far-red light increased leaf expansion (length and width) (Figure 1 and Figure 6; Appendix A). Increased leaf expansion is a typical shade acclimation response of lettuce that can be induced by a low ratio of red to far-red light [22,27,29]. The longer and wider leaves are also consistent with a shade avoidance response observed across a variety of other plant species [21,30,31]. Phytochromes play an essential role in controlling shade acclimation responses through the detection of the relative amount of far-red in the environment [32]. Phytochromes have two photo-reversible forms, Pr and Pfr, whose relative abundance depends on the light spectrum, and especially the red to far-red ratio [33]. Exposure to light with a high red to far-red ratio changes the protein structure of the phytochromes, converting Pr to Pfr, which reverts to Pr slowly in the dark or in response to light with a low red to far-red ratio. The Pfr form can be translocated to the nucleus to bind with phytochrome interacting factors (PIFs) and induce changes in gene expression [34,35]. Persistent far-red light application boosts the expression of genes involved in leaf elongation and expansion and increases gibberellin production [36].

The PPFD gradient also affected leaf morphology in the perpendicular light gradient study. Higher PPFD decreased leaf length in the early harvest and increased leaf width in the late harvest (Figure 6). There are contrasting reports of the effect of light intensity on lettuce leaf morphology, which suggests that lettuce has high phenotypic plasticity. Some studies report shorter and narrower leaves in response to higher PPFD [37,38]. Other studies show decreases in the length-to-width ratio in response to higher PPFD in lettuce [39], which would result from decreases in leaf length, increases in leaf width, or some combination of the two, as seen in our study. A higher daily light integral (DLI) has been linked to shorter lettuce leaves, as well as decreases in the length-to-width ratio, in a light intensity study with varying photoperiods and light quality [40]. Lettuce plants grown under higher light intensities also have been reported to have increased leaf width and length [41,42]. The inconsistency in the morphological effects of PPFD may be due to the genotypic differences among lettuce cultivars, but our results confirm that lettuce leaf morphology does respond to light quality and intensity.

### 3.2. Larger Leaves Lead to Increased Canopy Size

In the far-red gradient experiment, the longer and wider leaves in response to increased far-red light were correlated with increased total leaf area and PCS (Figure 2). Total leaf area and PCS were highly correlated with each other in both the far-red gradient study (Early harvest: *p <* 0.0001; *R* = 0.91; Late harvest: *p <* 0.0001; *R* = 0.89) and the perpendicular light gradient study (Early harvest: *p <* 0.0001; *R* = 0.96; Late harvest: *p <* 0.0001; *R* = 0.89). The percentage of ground cover, which is based on PCS, is correlated with radiation capture and carbon gain [28]. A larger PCS increases the amount of incident light and thus canopy photosynthesis and growth. This can be a self-re-enforcing process: plants that develop a larger PCS will be able to absorb more light, grow faster, and thus produce additional canopy faster than plants with a smaller PCS [43].

### 3.3. Larger Canopies Intercept More Light

Total leaf area and PCS at both harvests in both experiments were positively correlated with the cumulative amount of incident light received by plants. The cumulative incident light in turn was positively correlated with the dry weight of the plants (Figure 3 and Figure 8). Digital imaging has been used previously to determine canopy size and to estimate incident, intercepted, or absorbed light, resulting in a linear relationship between canopy size and incident light in multiple types of plants [28,44,45], including lettuce [46]. PCS has previously been shown to correlate with dry weight [47,48]. In the current study, the PCS at each harvest was positively correlated with cumulative incident light received by the plant and the shoot dry weight.

Leaf area was measured on the combined leaves from each plant. Due to the rosette structure of lettuce, there is inevitable overlap among leaves, which increases as the plant grows. We quantified the overlap ratio (total leaf area/PCS). In both studies, the overlap ratio at the early harvest was close to one (indicative of little or no overlap) and higher at the late harvest (Appendix A). The overlap reduces the amount of incident light per unit leaf area, due to intracanopy shading. This makes PCS a more appropriate parameter for the estimation of incident light than the total leaf area of a plant. Prior studies have found that the amount of light intercepted by plants increases in a curvilinear fashion in response to increasing leaf area [49,50,51]. However, PCS is directly correlated with the amount of incident light which a plant receives, regardless of growth stage.

### 3.4. Far-Red Light Increases Light Interception and Plant Biomass More Efficiently Than PPFD

In the far-red gradient study, far-red levels were positively correlated with the incident light and dry weight of the plant (Figure 4). Increasing far-red light increased leaf size, which increased total leaf area and projected canopy size. The larger canopy increased incident light, enabling the plants to grow larger. In lettuce, the amount of light received throughout the crop cycle has been shown to be linearly correlated with biomass production [52,53]. However, this study could not answer an important question: how does the efficacy of far-red light and PPFD compare in regard to driving biomass production? This question is important, because it determines whether it makes sense to replace some of the LEDs that provide PPFD with far-red LEDs in LED fixtures.

By examining the effect of far-red over a wide range of PPFDs, the perpendicular light gradient experiment allowed for quantification of the efficacy of PPFD and far-red light at increasing the incident light that plants receive. Across both harvests, the incident light increased with both increasing far-red and PPFD levels. At the early harvest, far-red photons were 57% more effective at increasing incident light than PPFD (Figure 9). This is consistent with a study that improved radiation capture by replacing blue light with far-red light, which increased the biomass of kale and lettuce due to increased leaf size [29]. Improving canopy size early in the plant’s growth and development is of benefit to growers, as seedlings capture little supplemental light due to their small size [54]. A larger canopy size allows plants to capture more light to drive growth, as light interception is a major determining factor for carbon gain. At the late harvest, far-red light was 183% more effective than PPFD light at increasing the cumulative incident light.

Both PPFD and far-red light were positively correlated with biomass at both harvests in the perpendicular light gradient study (Figure 9). Increasing PPFD typically increases the biomass of lettuce [41,55,56], although there is a PPFD threshold above which additional light decreases lettuce biomass [56,57]. In addition, high light intensity can increase disorders such as calcium deficiency and tipburn [55,58]. The recommended light level for lettuce under sole-source lighting is around 250 µmol m^−2^ s^−1^ when provided over a 16-h photoperiod [59]. The design of the perpendicular light gradient study allowed for comparison of the effectiveness of PPFD and far-red light at increasing biomass. Per incident photon, far-red light was 92.5% and 162% more effective than PPFD at increasing biomass at the early and late harvests, respectively. This suggests that replacing LEDs that provide PAR with far-red LEDs in LED fixtures can increase canopy size, incident light, and biomass, without the need to increase the total photon flux density. There must be a maximum threshold to the percentage of far-red light that can replace PAR in LED fixtures, without decreasing the LUE or causing adverse effects on plant quality. More investigation of these maximum levels of far-red light is needed. Shade tolerance and avoidance responses to far-red light vary among species and cultivars, requiring further experimentation to find the optimal amount of far-red light for different crops.

### 3.5. Light Use Efficiency

The LUE_PPFD_ was unaffected by far-red supplementation in the far-red gradient study, while the LUE_TOTAL_ decreased with increasing far-red light (Figure 5). Zou et al. found that increasing far-red light caused a decrease in leaf absorptance, because far-red light increased specific leaf area (SLA, leaf area/leaf dry weight), resulting in thinner leaves [27]. The lower leaf absorptance means that less light is available for photosynthesis and resulted in a decrease in leaf photosynthesis per unit leaf area in response to the long-term application of supplemental far-red light [27]. However, the increased leaf expansion in response to far-red light needs to be taken into consideration when considering canopy photosynthesis, as a larger canopy will intercept more light. The effect of far-red light on canopy photosynthesis thus depends on the opposing effects of lower photosynthesis per unit leaf area versus the increase in canopy size. In contrast to Zou et al. [27], we found no statistically significant association between far-red light levels and SLA in our studies, which could be due to the variety of lettuce that was used in the studies.

The LUE_PPFD_ in the perpendicular light gradient study increased with increasing far-red light, while the LUE_TOTAL_ was unaffected (Figure 10). The finding that the LUE_TOTAL_ was unaffected by the amount of far-red light suggests that far-red light was used as efficiently as PPFD to produce biomass. Our results agree with recent studies that show that far-red light is photosynthetically active when provided in combination with PPFD, because the Emerson effect results in synergistic increases in Φ_PSII_ [17,18,60]. As a result, far-red light increases gross canopy photosynthesis as efficiently as PPFD. Comparison of our LUE results with prior studies is difficult, because prior studies calculated LUE based on the amount of light provided to the growing space [27,61], rather than the amount of light reaching the canopy. Light use efficiency values, based on the amount of light provided to the growing space, range from 0.43 to 0.64 g mol^−1^ depending on the plant species [61]. Compared to other species, lettuce LUE was on the low end of the spectrum, with an LUE of 0.46 g mol^−1^. The edible dry matter LUE varies more drastically (0.16–0.44 g mol^−1^) among species, as the edible portion of plants can differ greatly. Lettuce was on the high end of this spectrum, with an LUE of 0.42 g mol^−1^ [61]. Zou et al. reported a substantially higher LUE (~0.7 g mol^−1^), based on light provided to the growing space, rather than reaching the crop. Their high LUE values may be the result of using transplants, thus eliminating the seedling stage when plants intercept little of the provided light. However, LUE values based on incident light on the growing space, rather than the canopy, are hard to compare since they depend on plant density. Using only the light incident on the plants, we found LUE_TOTAL_ values around 0.61 g mol^−1^ at the early harvest and 0.65 g mol^−1^ at the late harvest in the far-red gradient study. The perpendicular light gradient study resulted in larger LUE differences between the early and late harvests at 0.67 and 0.41 g mol^−1^, respectively. The lettuce LUE for lettuce reported by Wheeler et al. was 0.42 g mol^−1^ (based on shoot biomass) [61], lower than most of our values. This is not surprising, since they estimated LUE based on the total light provided to the growing space, while we used only the incident light reaching the canopy. However, when they estimated LUE using a densely spaced transplanting scheme, which would ensure that most light reaches the crop, LUE increased 66% to 0.74 g mol^−1^. While slightly higher than our LUE values, the difference may be due to their use of CO_2_ enrichment. 

The low LUE values from the late harvest of the perpendicular light gradient study may be related to the overlap ratio. The lettuce plants in the far-red gradient study had higher overlap ratios than the lettuce plants in the perpendicular gradient study (Appendix A). Far-red light is poorly absorbed by plant leaves, resulting in high amounts of far-red light being reflected and transmitted in plant canopies [62,63,64]. Far-red light can penetrate leaves and the canopy more deeply than red and blue light, which may excite photosynthetic machinery in cell layers that receive little energy from red and blue light, which are more readily absorbed in the upper layers of the leaf [65]. The higher overlap ratio allows leaves lower in the canopy to absorb transmitted far-red light, where it can contribute significantly to photosynthesis (S. Zhen, pers. comm.). Because a higher overlap ratio confers the ability of lower leaves to better absorb light transmitted by the upper canopy, it may increase the LUE. We did indeed see a trend towards higher LUE_TOTAL_ in plants with a larger overlap ratio in the late harvest of the perpendicular light gradient study (*p* = 0.01 and *R*^2^ = 0.20). However, the far-red gradient study did not display any significant trend between LUE and overlap ratio.

Both the LUE_TOTAL_ and LUE_PPFD_ were negatively correlated with SLA at both harvests in the perpendicular light gradient study. In the far-red gradient study, the LUE_TOTAL_ also was negatively correlated with SLA at both harvests, while LUE_PPFD_ was negatively correlated only at the late harvest (Figure 11). Specific leaf area is an important trait indicative of leaf acclimation to environmental conditions. Increasing the SLA allows for the plant to produce more leaf area, increasing incident light, but a high SLA lowers specific leaf nitrogen, chlorophyll content per unit leaf area, light absorptance, and photosynthetic rates per unit leaf area in lettuce [27]. While the relationship between SLA and LUE has not been well studied, specific leaf nitrogen (g of N per unit leaf area) is negatively correlated with SLA [66], while high LUE has been linked to high specific leaf nitrogen in a variety of crops [67,68,69,70]. The connection between a low SLA and high specific leaf nitrogen, combined with the positive correlation between specific leaf nitrogen and LUE, suggests that plants with a lower SLA will have a higher LUE, consistent with our findings (Figure 11). Plants with a low SLA will likely be more efficient at absorbing light and converting that light into biomass, but the tradeoff is a reduction in canopy size and incident light.

### 3.6. Implications

Using the data from the perpendicular light gradient study, we compared the efficiency of far-red light and PPFD related to light interception (morphology), light use efficiency (physiology), and biomass accumulation. Far-red light was more efficient than PPFD at increasing the PCS, incident light, and biomass of lettuce, while also increasing LUE_PPFD_ but not LUE_TOTAL_. The effects on growth were more pronounced at the late harvest, likely because the effects accumulate over time. Therefore, replacing some of the LEDs emitting in the PAR range with far-red LEDs from 700 to 735 nm should provide benefits to growers using sole-source lighting, by increasing canopy size and LUE_PPFD_. Far-red LEDs commonly have an emission peak close to 730 nm, which is effective at inducing phytochrome-mediated responses [71], as well as increasing Φ**_PSII_** [17]. The combined physiological and morphological effects of far-red light, when combined with PPFD, can result in larger lettuce plants grown at a lower electricity cost. Different waveband LEDs vary in their efficacy (photons per joule), which should be taken into consideration in LED fixture design. Far-red LEDs have higher efficacy than blue, green, red, cool-white, and warm-white LEDs [72]. The inclusion of far-red LEDs will increase the fixture efficacy if defined as total photon flux per Joule. Unfortunately, fixture efficacy is often defined as photosynthetic photon flux (400–700 nm) per Joule [73], in which case the inclusion of far-red LEDs would lower efficacy. Given the clear evidence that far-red photons are photosynthetically active [17,18,60] and increase biomass production, the standards for measuring efficacy may need to be revised.

It is important to consider that too much far-red light may negatively impact quality and growth. Future studies are needed to establish the best ratios of far-red light to PPFD. The maximum observed PPFD to far-red ratios ranged from 40.4 µmol m^−2^ s^−1^ of PPFD for every 1 µmol m^−2^ s^−1^ of far-red light to 5.17 µmol m^−2^ s^−1^ of PPFD for every 1 µmol m^−2^ s^−1^ of far-red light, similar to the PPFD to far-red photon ratio in sunlight (5.3) [18]. The morphological and growth responses observed in the studies were linear across the ranges of PPFD and far-red light used. While lettuce still benefited from PPFD to far-red ratios that were similar to sunlight, the effect of far-red light may vary among species and cultivars, which necessitates studies on other crops. It is also important to consider that these studies were conducted in a sole-source lighting environment and may not be applicable to greenhouse conditions, where sunlight can provide a significant amount of far-red light. However, for sole-source lighting, our results suggest that lights with PPFD to far-red ratios that are similar to that in sunlight increase growth compared to lights with a smaller fraction of far-red light.

We used digital imaging to enable the calculation of LUE. Bi-weekly PCS measurements were used to estimate the daily PCS. Combined with the instantaneous photon flux density, the cumulative incident light received by the plants and LUE were calculated. The method is simple and does not require expensive equipment. It allows for separation of treatment effects that impact incident light versus how efficiently that light is used to produce biomass. It can be used to quantify the effects of a wide range of production practices on crop growth, such as light spectrum and intensity, CO_2_ concentration, fertilization, temperature, and irrigation practices. In addition, it provides a simple way to gain a better understanding of growth differences among different cultivars and species.

### 3.7. Conclusions

Supplemental far-red light has profound effects on lettuce morphology and growth. Far-red light increases leaf expansion and canopy size, which causes increases in the cumulative incident light and biomass. Far-red photons are more effective at increasing incident light than photons in the 400–700 nm region. The impact of far-red light was greater later in the growing cycle, probably due to the cumulative effect over time. Plants in our study did not overlap each other, and the effects of far-red light may diminish when plants compete for light. While the effect of far-red on LUE differed between the two studies, the perpendicular light gradient study indicates that far-red light is more effective at producing biomass than light from the traditional PAR range (400–700 nm). Including far-red light in LED fixtures for sole-source lighting, to provide a ratio of PAR to far-red light that is similar to that of sunlight, can promote lettuce growth. The growth response observed to supplemental far-red light was linear up to the observed at the experiment maximum of 30 µmol m^−2^ s^−1^ of far-red light, and detrimental effects on quality were not observed.

## 4. Materials and Methods 

### 4.1. Growth Chamber Conditions

Two studies were conducted using the same 0.8 m × 1.8 m growth chamber (E15, Conviron, Winnipeg, MB, Canada). The first study used a far-red gradient combined with a uniform PPFD, while the second study used perpendicular far-red and PPFD gradients. Cool-white LED panels (Cool white 225 LED ultrathin grow light panel, Yescom USA, City of Industry, CA, USA) were hung 0.6 m above the floor of the growth chamber to provide the PPFD, while the far-red light was provided using custom-built bars with far-red LEDs (peak at 735 nm with a full width at half maximum of 25 nm). These LEDs were selected because they can induce the Emerson enhancement effect [60], while LEDs with a similar spectrum were previously shown to induce phytochrome-like responses as well [21].

The fractions of the wavebands of the white LED light were 0.39 blue (400–500 nm), 0.40 green (500–600 nm), 0.19 red (600–700 nm), and 0.02 far-red (700–800 nm) (see Appendix A for full spectra of the white and far-red LEDs). The light spectrum was unique for each plant because of the different ration of white and far-red light and was measured at the location of each individual plant prior to the start of the experiment at the height of the seedling canopy to determine the PPFD and far-red light intensity that each plant was exposed to. These measurements were taken using a spectroradiometer (SS-110, Apogee Instruments, Logan, UT, USA). Lights were on an 18 h on/6 h off schedule.

The far-red gradient study created the gradient by positioning two custom-built LED bars, with 4 far-red LEDs each, on one side of the growth chamber. PPFD was relatively uniform (207 ± 13 µmol m^−2^ s^−1^, mean ± SD) and provided by eight evenly spaced cool-white LED panels. Far-red light levels (700–800 nm) ranged from 4.9 to 28.0 µmol m^−2^ s^−1^. For the far-red gradient study, the temperature was 21.8 ± 1.4 °C with a vapor pressure deficit of 1.6 ± 0.2 kPa. The CO_2_ concentration was the same as that of the ambient air.

The perpendicular light gradient study established far-red and PPFD gradients orthogonally to each other to give a range of PPFD and far-red combinations. Eight cool-white LED panels were positioned to create a PPFD gradient by skewing them towards one side of the growth chamber, creating a PPFD gradient from 111 to 245 µmol m^−2^ s^−1^. Far-red was provided from the back side of the growth chamber, using a custom far-red LED light bar created with 16 far-red LEDs (peak at 733 nm). An aluminum reflector was installed over the far-red LEDs to create a larger gradient. Far-red light levels ranged from 4.7 to 32.8 µmol m^−2^ s^−1^. For the perpendicular light gradient study, the temperature was maintained at 24.4 ± 2.9 °C (mean ± SD) with a vapor pressure deficit of 1.4 ± 0.4 kPa. The CO_2_ concentration was the same as that of the ambient air.

### 4.2. Plant Material

For each study, three “Green Salad Bowl” lettuce seeds (Seedway, Hall, NY, USA) were sown into 60 10-cm square pots (42 plants/m^2^) filled with soilless substrate (SS#1-F1P Soilless Potting Mix, SunGro, Agawam, MA, USA). Plants were thinned to one plant per pot at 5 days after sowing, selecting the most uniform seedlings. Plants were fertigated using 100 mg L^−1^ N water-soluble fertilizer solution (15N-2.2P-12.5K, 15-5-15 Calcium + Magnesium LX; JR Peters Inc., Allentown, PA, USA) as needed, approximately three times a week. To prevent algae growth on the substrate surface and facilitate imaging, a mixture of H_2_O_2_ and peroxyacetic acid (ZeroTol 2.0, Biosafe Systems, East Hartford, CT, USA) was sprayed on the surface of the substrate every 7 days. A 1:500 dilution of ZeroTol 2.0 to water was used for the first spray treatment and the concentration was increased to 1:200 for subsequent treatments.

### 4.3. Digital Imaging and Image Analysis

During both studies, plants were digitally imaged twice weekly using a multispectral digital imaging system (TopView, Aris, Eindhoven, The Netherlands) beginning 7 days after sowing. During the studies, plants were imaged a total of eight times to quantify the projected canopy size of each plant. Although the imaging system takes monochrome pictures under different colors of LEDs, only the image of chlorophyll fluorescence was used. Chlorophyll fluorescence imaging simplifies the image analysis, by making it easy to separate the canopy from other pixels in the image. Using a band-pass filter (>695 nm) in front of a monochrome camera and exciting the crop with blue actinic light (peak at 450 nm) in a light-secure chamber, the only thing visible in the image is the fluorescence emitted by chlorophyll (Appendix A).

To calculate the PCS, chlorophyll fluorescence images were analyzed in Fiji (Schindelin et al., 2012; www.fiji.sc). First, a manually set intensity threshold was applied to each picture to remove as much background as possible. A mask was created from the thresholded image, which was then used to calculate the number of pixels for the PCS of each plant. A scale was included in each picture to allow for conversion from number of pixels to the actual PCS.

### 4.4. Harvest

In both studies, plants were close to overlapping 16 days after germination, and half of the plants were harvested. The remaining plants were harvested 25 days after germination as plants were again becoming close to overlapping. At both harvests, the longest leaf from each individual plant was measured for both length and width. The total leaf area of all leaves from a plant was measured using a leaf area meter (LI-3100, Li-Cor, Lincoln, NE, USA). The leaf overlap ratio was determined by dividing the total leaf area by the projected canopy size at the end of each study. Shoot plant material was dried at 80 °C for three days, and dry weight was recorded. Specific leaf area was calculated as the total leaf area of a plant divided by the shoot dry weight.

### 4.5. Modeling of Projected Canopy Size to Calculate Light Use Efficiency

The PCS data from the bi-weekly chlorophyll fluorescence images were log transformed, after which a quadratic equation was fitted to describe the PCS of each plant over time (Excel, Version 2002, Microsoft, Seattle, WA, USA). The equations had a *R*^2^ > 0.99. Using these equations, the PCS for every plant was estimated for each day. Utilizing the PCS, PPFD, and far-red intensity for each plant, incident photon flux for each day was calculated by multiplying the projected canopy size by PPFD and (PPFD + far-red intensity). Cumulative incident photon flux was calculated by summing those daily values. The shoot dry weight was divided by the cumulative incident photon flux to calculate the light use efficiency (LUE; g of biomass/mol of light) from both 400–700 nm (LUE_PPFD_) and 400–800 nm (LUE_TOTAL_), as a measure of how efficiently incident photon flux is used to produce biomass.

For statistical analysis, all data were analyzed using JMP Pro (version 15.0.0) using generalized linear models using a backwards stepwise selection to check for significance of FR, PPFD, and interactions between the two.

## Figures and Tables

**Figure 1 plants-10-00166-f001:**
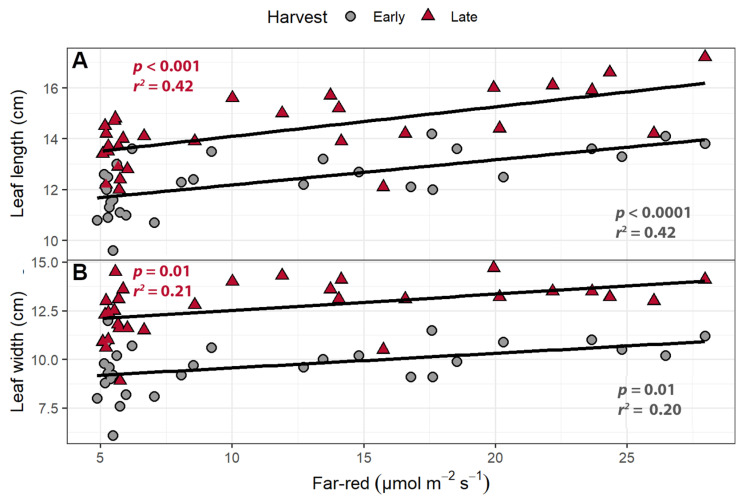
(**A**) The relationship between supplemental far-red light intensity and leaf length and (**B**) leaf width of “Green Salad Bowl” lettuce. Plants were grown under a photosynthetic photon flux density of 207 ± 13 µmol m^−2^ s^−1^ supplemented with varying amounts of far-red. The early harvest occurred 16 days after germination, while the late harvest occurred 25 days after germination.

**Figure 2 plants-10-00166-f002:**
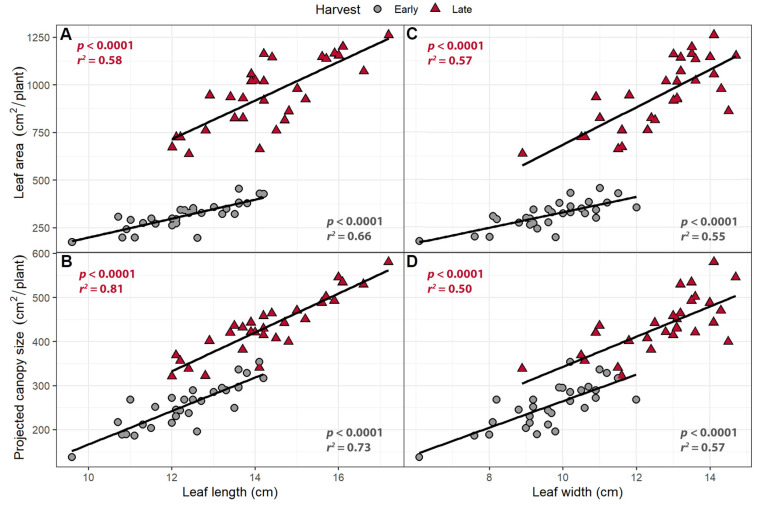
(**A**) The correlation between (**A**) leaf length and leaf area; (**B**) leaf length and projected canopy size; (**C**) leaf width and leaf area, and (**D**) leaf width and projected canopy size. “Green Salad Bowl” lettuce plants were grown under a photosynthetic photon flux density of 207 ± 13 µmol m^−2^ s^−1^ supplemented with varying amounts of far-red. The early harvest occurred 16 days after germination, while the late harvest occurred 25 days after germination.

**Figure 3 plants-10-00166-f003:**
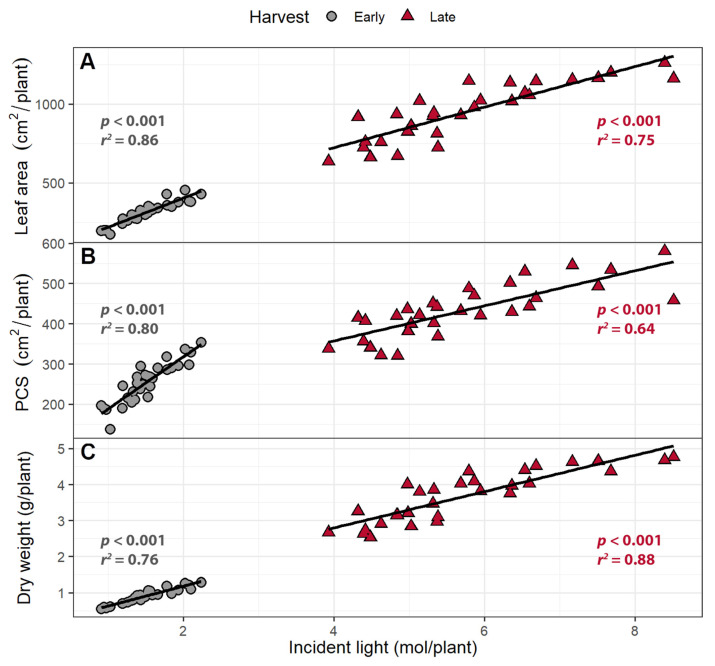
The correlation between (**A**) total leaf area and (**B**) projected canopy size (PCS) and total cumulative incident light (400–800 nm) over the growing period; (**C**) The correlation between incident light and shoot dry weight. Plants were grown under a photosynthetic photon flux density of 207 ± 13 µmol m^−2^ s^−1^ supplemented with varying amounts of far-red. The early harvest occurred 16 days after germination, while the late harvest occurred 25 days after germination.

**Figure 4 plants-10-00166-f004:**
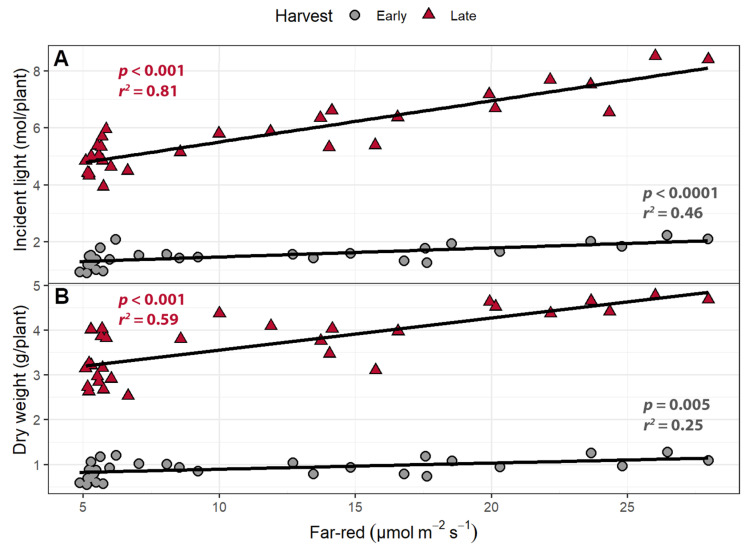
(**A**) The correlation between supplemental far-red light intensity and cumulative incident light (400–800 nm) and (**B**) dry weight. “Green Salad Bowl” lettuce (*Lactuca sativa*) plants were grown under a photosynthetic photon flux density of 207 ± 13 µmol m^−2^ s^−1^ supplemented with varying intensities of far-red. The early harvest occurred 16 days after germination, while the late harvest occurred 25 days after germination.

**Figure 5 plants-10-00166-f005:**
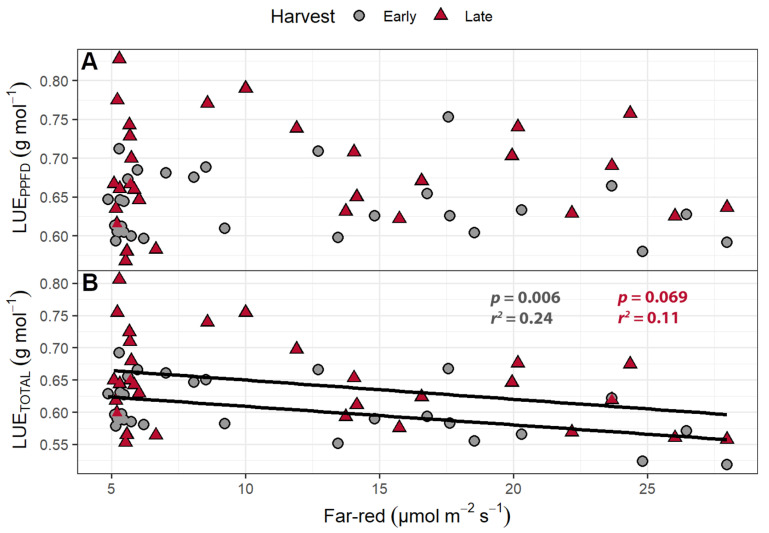
(**A**) The effect of supplemental far-red light on light use efficiency calculated based on photosynthetic photon flux density (LUE_PPFD_) and (**B**) total photon flux density (LUE_TOTAL_). The LUE_PPFD_ and LUE_TOTAL_ were calculated by dividing the shoot dry weight by the cumulative incident photosynthetic photon flux density (PPFD; 400–700 nm) or the incident total photon flux (400–800 nm), respectively, to determine how efficiently plants used incident light to produce biomass. “Green Salad Bowl” lettuce (*Lactuca sativa*) were grown under a PPFD of 207 ± 13 µmol m^−2^ s^−1^ supplemented with varying amounts of far-red. The early harvest occurred 16 days after germination, while the late harvest occurred 25 days after germination.

**Figure 6 plants-10-00166-f006:**
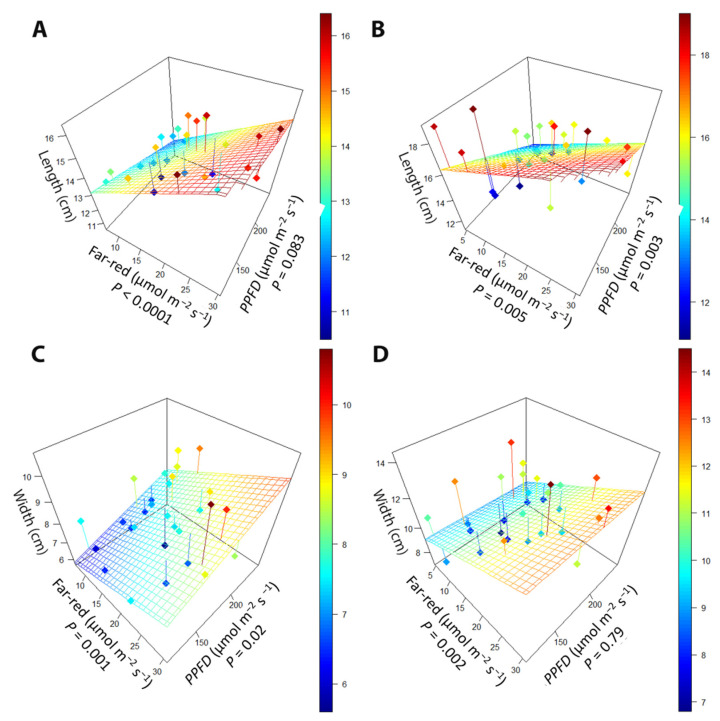
The relationship between supplemental far-red light, photosynthetic photon flux density (PPFD), and (**A**) leaf length at the early harvest, (**B**) leaf length at the late harvest, (**C**) leaf width at the early harvest, and (**D**) leaf width at the late harvest. “Green Salad Bowl” lettuce (*Lactuca sativa*) plants were grown under photosynthetic photon flux densities ranging from 111 to 245 µmol m^−2^ s^−1^ and a far-red gradient ranging from 4.7–32.8 µmol m^−2^ s^−1^. The early harvest occurred 16 days after germination, while the late harvest occurred 25 days after germination.

**Figure 7 plants-10-00166-f007:**
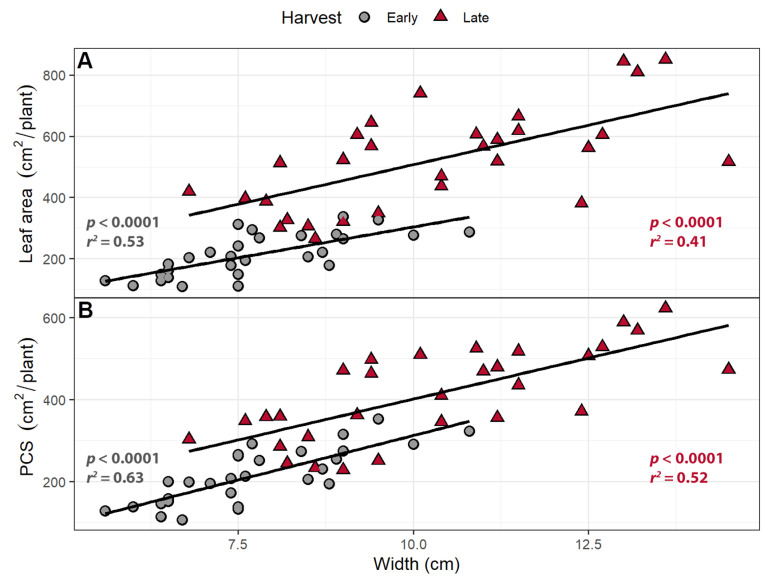
The relationship between leaf width and (**A**) leaf area and (**B**) projected canopy size (PCS). “Green Salad Bowl” lettuce (*Lactuca sativa*) were grown under a photosynthetic photon flux density ranging from 111 to 245 µmol m^−2^ s^−1^ supplemented with a far-red gradient which ranged from 4.7 to 32.8 µmol m^−2^ s^−1^. The early harvest occurred 16 days after germination, while the late harvest occurred 25 days after germination.

**Figure 8 plants-10-00166-f008:**
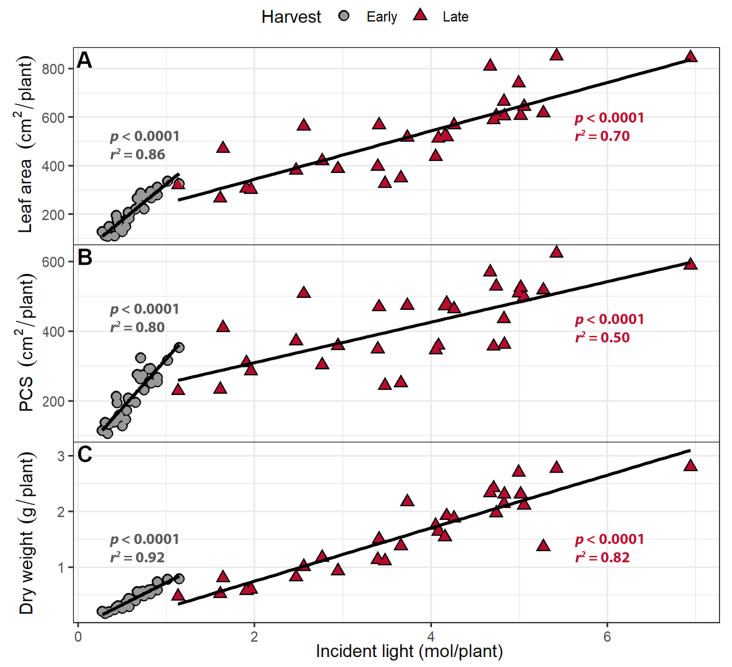
(**A**) The correlation between cumulative incident light (400–800 nm) and leaf area, (**B**) projected canopy size, and (**C**) shoot dry weight. “Green Salad Bowl” lettuce (*Lactuca sativa*) plants were grown under photosynthetic photon flux densities ranging from 111 to 245 µmol m^−2^ s^−1^ and a perpendicular far-red gradient which ranged from 4.7 to 32.8 µmol m^−2^ s^−1^. The early harvest occurred 16 days after germination, while the late harvest occurred 25 days after germination.

**Figure 9 plants-10-00166-f009:**
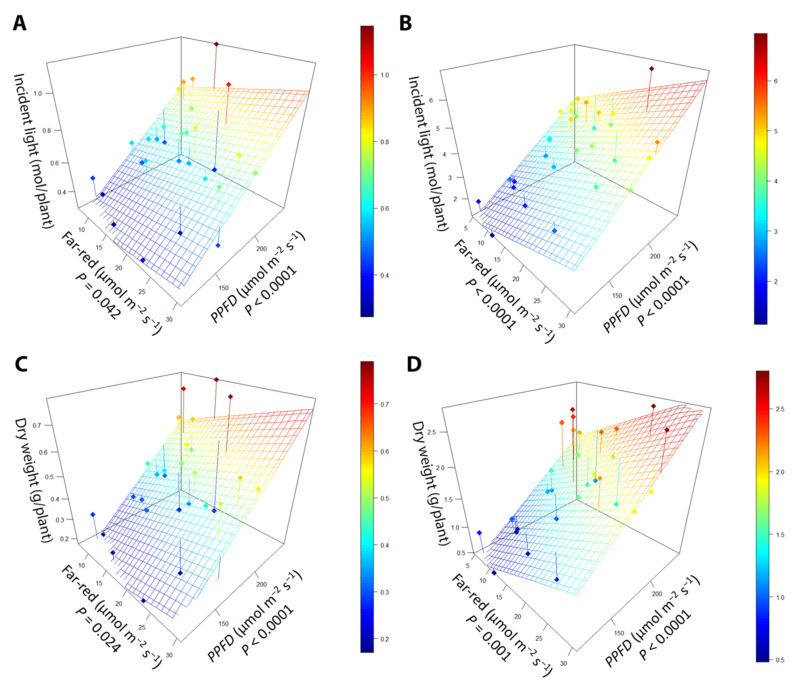
The relationship between supplemental far-red light intensity, photosynthetic photon flux density (PPFD), and incident light at (**A**) the early harvest and (**B**) the late harvest. The relationship between far-red light intensity, PPFD, and dry weight in (**C**) the early harvest and (**D**) the late harvest. “Green Salad Bowl” lettuce (*Lactuca sativa*) plants were grown under a PPFD ranging from 111 to 245 µmol m^−2^ s^−1^ and a perpendicular far-red gradient ranging from 4.7 to 32.8 µmol m^−2^ s^−1^. The early harvest occurred 16 days after germination, while the late harvest occurred 25 days after germination.

**Figure 10 plants-10-00166-f010:**
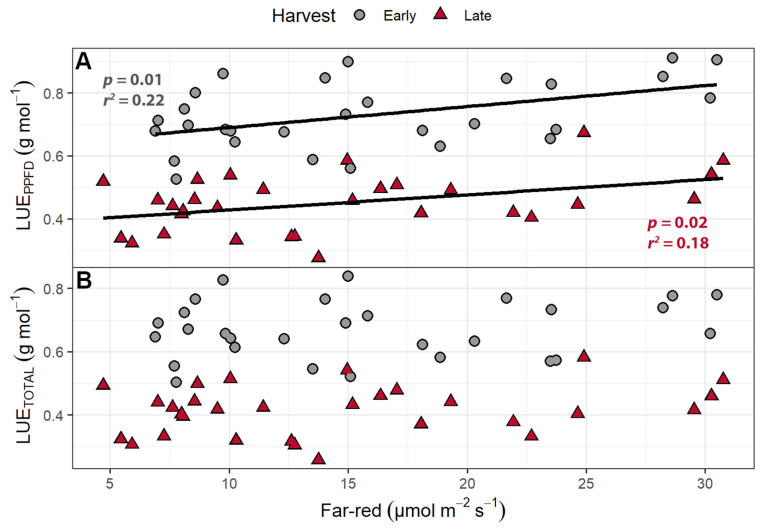
(**A**) The effect of supplemental far-red light on light use efficiency calculated based on photosynthetic photon flux density (PPFD) (LUE_PPFD_) and (**B**) total photon flux density (LUE_TOTAL_) of “Green Salad Bowl” lettuce (*Lactuca sativa*). Plants were grown under a PPFD ranging from 111 to 245 µmol m^−2^ s^−1^ with a perpendicular far-red light gradient ranging from 4.7 to 32.8 µmol m^−2^ s^−1^. The LUE_PPFD_ and LUE_TOTAL_ were calculated by dividing the shoot dry weight by the cumulative incident light from 400 to 700 nm or 400 to 800 nm, respectively, to determine how effectively plants used incident light to produce biomass. The early harvest occurred 16 days after germination, while the late harvest occurred 25 days after germination.

**Figure 11 plants-10-00166-f011:**
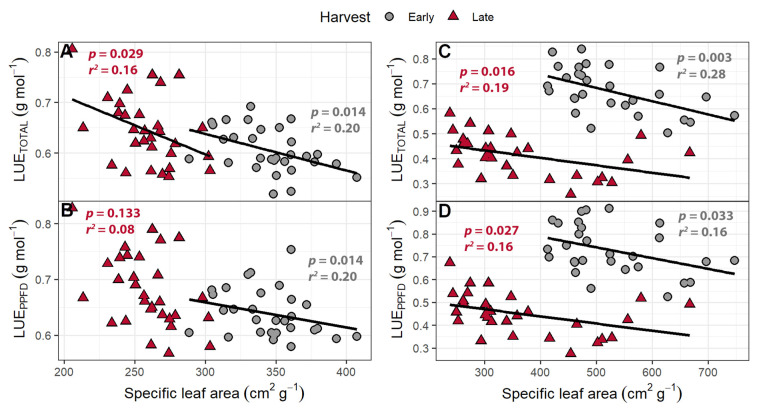
(**A**,**C**) The correlation between specific leaf area and light use efficiency of “Green Salad Bowl” lettuce (*Lactuca sativa*) calculated based on total photon flux density (LUE_TOTAL_) and (**B**,**D**) photosynthetic photon flux density (LUE_PPFD_). Plants were grown with 207 ± 13 µmol m^−2^ s^−1^ of white light and 4.9 to 28.0 µmol m^−2^ s^−1^ far-red light (**A**,**B**) or 111 to 245 µmol m^−2^ s^−1^ of white light and 4.7 to 32.8 µmol m^−2^ s^−1^ far-red light (**C**,**D**). Plants were harvested after 16 (early) or 25 days after germination (late).

**Table 1 plants-10-00166-t001:** Regression equation intercepts and coefficients for dry weight (g/plant), cumulative incident light (mol/plant), leaf length (cm), and leaf width (cm) for the early and late harvests, as a function of the far-red (FR) light level (µmol m^−2^ s^−1^) and photosynthetic photon flux density (PPFD, µmol m^−2^ s^−1^).

Trait	Harvest	Intercept	FR	PPFD	*R* ^2^
Dry weight	Early	–0.49	0.00792	0.00414	0.53
Dry weight	Late	–1.28	0.04276	0.01183	0.56
Incident light	Early	–0.50	0.00815	0.00520	0.58
Incident light	Late	–2.72	0.07946	0.02807	0.74
Length	Early	13.68	0.1478	–0.0126	0.57
Length	Late	19.40	0.1305	–0.0315	0.42
Width	Early	3.276	0.0966	0.0152	0.37
Width	Late	8.774	0.1385	–0.0025	0.29

## Data Availability

The data presented in this study are available at https://tinyurl.com/yy2mjknd.

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
