# Peer review of "Supplemental Far-Red Light Stimulates Lettuce Growth: Disentangling Morphological and Physiological Effects"

_plants, 2021, doi:10.3390/plants10010166_

Round 1

Reviewer 1 Report

The work concerns the possibility of far-red light supplementing in the closed cultivation of lettuce seedlings. The subject matter is reasoned and  presents an interesting problem. The authors made many detailed measurements, the results are showed in many interesting ways and the paper was prepared very carefully.

I have a few comments from the reviewed work.

  1. White light should be characterized in more detail (spectral composition) because the proportion of red and far-red in the spectrum can have a significant impact on the results. Additionally the amount of blue, green and yellow light also could be important, especially in the second experiment. Authors should take these factors into account when discussing the results also that they are different for different combinations in both experiments. They themselves emphasize in the discussion that these proportions are important.
  2. The whole paper is too expatiatory. Especially the introduction and discussion need to be shortened. Some elements from the introduction should be transferred to the methodology, and some, after shortening (marked in yellow) at the end of the introduction, as an element of the work objective. Some elements are so obvious that authors may omit them.
  3. In the first experiment, the control light should be white light with an increasing PFD value corresponding to the amount of FR added. It will be easier to answer what was more important - increasing sum of PFD or adding FR.
  4. The results are re-discussed too extensively in the discussion. The chapter 3.1 of discussion doesn't explain anything about work.
  5. Both the LUETOTAL and LUEPPFD were negatively correlated with SLA, this begs the question of the quality of the leaves. Especially that the cultivation lasted only 25 days (it is not known why only so much) and it is not known how these plants would grow further - in the ground, in a greenhouse or in closed systems.

With such extensive experiment, a clear summary would be useful at the end. What was positive, what was negative, which requires explanation and suggestions for practice.

Author Response

Reviewer #1:

  1. White light should be characterized in more detail (spectral composition) because the proportion of red and far-red in the spectrum can have a significant impact on the results. Additionally, the amount of blue, green and yellow light also could be important, especially in the second experiment. Authors should take these factors into account when discussing the results also that they are different for different combinations in both experiments. They themselves emphasize in the discussion that these proportions are important

We have included the spectrum of the white and far-red LED lights as supplemental figure 3 as well as the addition of more information to the materials and methods that better describes the lighting conditions (lines 523-524). This also includes the far-red spectrum as well.

  1. The whole paper is too expatiatory. Especially the introduction and discussion need to be shortened. Some elements from the introduction should be transferred to the methodology, and some, after shortening (marked in yellow) at the end of the introduction, as an element of the work objective. Some elements are so obvious that authors may omit them.

We have edited both the introduction and discussion to make it more concise.

  1. In the first experiment, the control light should be white light with an increasing PFD value corresponding to the amount of FR added. It will be easier to answer what was more important - increasing sum of PFD or adding FR.

For comment 3, the reviewer is proposing a different experiment than what we did and one that would be very challenging to set up (from a technical perspective). Our objective in the first experiment, as spelled out in the introduction, was to quantify the effect of supplemental far-red light. The 2nd study was able to answer the question of the relative efficacy of white and far-red light.

  1. The results are re-discussed too extensively in the discussion. The chapter 3.1 of discussion doesn't explain anything about work

Section 3.1 explains the phytochrome response to far-red light, which is integral to why and how plants are responding to far-red light and is essential to the discussion. We have edited the discussion to make it more concise.

  1. Both the LUETOTAL and LUEPPFD were negatively correlated with SLA, this begs the question of the quality of the leaves. Especially that the cultivation lasted only 25 days (it is not known why only so much) and it is not known how these plants would grow further - in the ground, in a greenhouse or in closed systems

Comment 5 was addressed by adding a few brief sentences to the materials and methods explaining why we conducted our harvests when we did (lines 572-574). The discussion also includes an in depth look at the relationship between LUE and SLA (lines 436-449).

With such extensive experiment, a clear summary would be useful at the end. What was positive, what was negative, which requires explanation and suggestions for practice.

Section 3.6 discusses the implications of this work, while section 3.7 summarizes the key conclusions.

Reviewer 2 Report

I think this is a very interesting paper on the effect of far-red light on lettuce growth. I like the method using chlorophyll fluorescence technique to measure projected canopy size. A topic that needs more discussion both in the introduction and in the discussion is differences in Emerson enhancement effects mediated by absorption of light in PSI, and the shade avoidance phytochrome effects.

As Zhen and Bugbee (2020) has shown, the Emerson enhancement effect decreases from 700 to 730 nm, and it probably completely stops between 730 and 750 nm. This is what is expected because absorption of light in the leaf (and also in PSI) declines steeply from 700 to zero at 750 nm. You have used far-red LED with peak at 733 nm with a full width at half maximum of 26 nm. The Emerson enhancement effect from these LEDs is therefore expected to be quite low. The growth stimulation effects are therefore probably mainly morphogenetic phytochrome effects.

Therefore, the spectrum of the far-red light used is very important and it should be more discussed. The addition of far-red light to LED light in growth chambers is mainly done with far-red LEDs with peaks around 730 nm because of its effect on phytochrome. If a growth stimulation (Emerson enhancement effect) of far-red light is important, then far-red LED with slightly shorter peak wavelengths (perhaps around 710 nm) may be better as long as the phytochrome effect is not much changed.

Another aspect of using additional far-red light is spectral light distribution within the leaves. Far-red light will be transmitted to deeper layers within the leaf (as green light). Therefore, it may be transmitted to deeper layers within the leaf which are not light saturated, and it will then increase photosynthesis more than in the upper layer which is already light saturated with blue and red light. Adding far-red light may therefore have a similar effect as adding green light (see Terashima et al. 2009, Plant Cell Physiol. 50(4): 684–697).

Author Response

Reviewer #2:

I think this is a very interesting paper on the effect of far-red light on lettuce growth. I like the method using chlorophyll fluorescence technique to measure projected canopy size. A topic that needs more discussion both in the introduction and in the discussion is differences in Emerson enhancement effects mediated by absorption of light in PSI, and the shade avoidance phytochrome effects,

The reviewer asked for more information about the Emerson enhancement effect in the introduction and discussion. While we already mentioned it in the introduction, we found we were lacking in the discussion section, and added a few lines and citations from 398-401.

As Zhen and Bugbee (2020) has shown, the Emerson enhancement effect decreases from 700 to 730 nm, and it probably completely stops between 730 and 750 nm. This is what is expected because absorption of light in the leaf (and also in PSI) declines steeply from 700 to zero at 750 nm. You have used far-red LED with peak at 733 nm with a full width at half maximum of 26 nm. The Emerson enhancement effect from these LEDs is therefore expected to be quite low. The growth stimulation effects are therefore probably mainly morphogenetic phytochrome effects.

Therefore, the spectrum of the far-red light used is very important and it should be more discussed. The addition of far-red light to LED light in growth chambers is mainly done with far-red LEDs with peaks around 730 nm because of its effect on phytochrome. If a growth stimulation (Emerson enhancement effect) of far-red light is important, then far-red LED with slightly shorter peak wavelengths (perhaps around 710 nm) may be better as long as the phytochrome effect is not much changed.

Shuyang Zhen’s work on far-red and the Emerson effect started in our lab, using far-red LEDs with a very similar spectrum to those used in the current study (Zhen and van Iersel, 2017). This spectrum induced a substantial Emerson effect, likely because wavelengths up to at least 732 nm are very effective (Zhen et al, 2019). Zhen and Bugbee (2020) used LEDs with a fairly wide spectrum, while Zhen et al (2019) used lasers with a very narrow spectrum, thus better able to determine which wavelengths are effective in inducing the Emerson effect.

Another aspect of using additional far-red light is spectral light distribution within the leaves. Far-red light will be transmitted to deeper layers within the leaf (as green light). Therefore, it may be transmitted to deeper layers within the leaf which are not light saturated, and it will then increase photosynthesis more than in the upper layer which is already light saturated with blue and red light. Adding far-red light may therefore have a similar effect as adding green light (see Terashima et al. 2009, Plant Cell Physiol. 50(4): 684–697).

We agree and are actually collaborating with Shuyang Zhen on this very topic. Not only can far-red light penetrate deeper into leaves than PAR, it reaches deeper into the canopy and can have a surprisingly strong impact on photosynthesis of lower canopy leaves. See lines 426-429.

Reviewer 3 Report

The manuscript presents valuable, complex, new data, however, in my opinion is could be shortened especially the discussion to avoid unnecessary redundancy. In my opinion, mixing of information of material and method within the results (e.g. l. 142 -149, l. 225-231) should be avoided.

Otherwise, presentation of more information regarding light conditions as spectrum of LED lamps should be added as well as calculations of LUE based on PAR and total light intensity. The comparison of these calculations with here used LUE based on incident light would contribute to improve scientific approaches to this topic. Moreover, the R:FR ratio seems to be required to understand a probable shade avoidance effect. More details for the light values for the perpendicular light gradient experiment are required. The figures 6 and 9 provide this information only roughly. However, I assume the mechanism of FR effects cannot be understood based on the presented experiments because no physiological investigations as measurement of gas exchange, net photosynthesis and so on were done. Especially if FR really contribute to photosynthesis, more physiological research is required. Nevertheless, the data are provide valuable arguments for adding FR to LED lamps.

It should be added who many plants per light situation were evaluated! The low r2 values is some figures presenting FR relationships could be probably increased by more replicats. It is also unclear whether there was a replication of the experiments also this is necessary for the scientific reliability.

In figure 1 and 5 there is an accumulation of dots around 5 µmol m-2 s-1 FR, why?

Some more remarks are annotated to the manuscript.

Author Response

The manuscript presents valuable, complex, new data, however, in my opinion is could be shortened especially the discussion to avoid unnecessary redundancy. In my opinion, mixing of information of material and method within the results (e.g. l. 142 -149, l. 225-231) should be avoided.

In response to the mixing of materials and methods in the results, we believe that a brief description of the studies in the results section can save the reader unnecessary confusion. Given that the Plants format requires the Results section to come before the Materials and Methods, it is important to give readers a brief intro into the two studies we conducted. Without that info, the results would be very hard to interpret.

Otherwise, presentation of more information regarding light conditions as spectrum of LED lamps should be added as well as calculations of LUE based on PAR and total light intensity. The comparison of these calculations with here used LUE based on incident light would contribute to improve scientific approaches to this topic. Moreover, the R:FR ratio seems to be required to understand a probable shade avoidance effect. More details for the light values for the perpendicular light gradient experiment are required. The figures 6 and 9 provide this information only roughly. However, I assume the mechanism of FR effects cannot be understood based on the presented experiments because no physiological investigations as measurement of gas exchange, net photosynthesis and so on were done. Especially if FR really contribute to photosynthesis, more physiological research is required. Nevertheless, the data are provide valuable arguments for adding FR to LED lamps.

Spectrum of the LEDs is addressed by supplemental figure 3, while supplemental figures 4 & 5 show the light distribution in the perpendicular gradient experiment. Calculation of R:FR ratio may be useful in some respects but requires complete rework of the discussion and introduction as well as all of the figures. We believe that the way the information was analyzed and presented provides a unique ability to compare the efficacy of PPFD vs supplemental far-red light.

It should be added who many plants per light situation were evaluated! The low r2 values is some figures presenting FR relationships could be probably increased by more replicats. It is also unclear whether there was a replication of the experiments also this is necessary for the scientific reliability.

Additional information was added in the materials and methods (line 525) clarifying that each plant received a unique lighting treatment. Because each plant was exposed to a slightly different light environment, there are no replicated treatment levels in these studies. The experimental unit was a single plant, and the light environment of each plant was quantified. Using single plants as the experimental unit does indeed result in significant variability, explaining the relatively low r2 values. But it also greatly increases the total number of experimental units that can be used for data analysis, and thus the degrees of freedom and statistical power. That in turn resulted in low P-values, despite relatively low R2 values.

In figure 1 and 5 there is an accumulation of dots around 5 µmol m-2 s-1 FR, why?

                 There was a relatively large number of plants that received low levels of far-red light due to the plants furthest from the light bars receiving similarly low levels of far-red.

The reviewer questioned why we introduced a new figure (11) in the discussion

The purpose of figure 11 is to help discuss reasons for differences in LUE. The graph is not directly related to our treatments, but rather helps to explain observed differences in LUE. We therefore feel it better fits in the discussion than in the results.

Round 2

Reviewer 1 Report

The article has been greatly improved, especially the introduction. I further think that there is too much discussion of own results in the discussion, and you should definitely not include figures of your own results in this chapter.
The spectrum of white light (supplemental figure 3) shows that it had little amount of red light compared to blue light, hence perhaps such a strong reaction of plants to far-red light.
Nevertheless, the work contains very interesting results that are worth publishing.

Author Response

In response to the reviewer's comment to not introduce new figures in the discussion, we now refer to the relationship between LUE and SLA twice in the Results section (lines 191-192 and 279-281, where we also refer to Figure 11).

We agree that there is substantial discussion of our results in the discussion section, but that is necessary to place our data in the context of what is already in the literature.

Reviewer 3 Report

The revised paper enables the reader especially by adding more information regarding light conditions to understand the experimental design, to follow the argumentations of the authors and to compare this research to others.

There are few typing mistakes to correct before final publication (e.g. line 411).

Author Response

We read through the manuscript once and fixed a few more typos/minor grammar issues.